



# Eddy Covariance Measurements Highlight Sources of Nitrogen Oxide Emissions Missing from Inventories for Central London

Will S. Drysdale[1], Adam R. Vaughan[1], Freya A. Squires[1,a], Sam J. Cliff[1], Stefan Metzger[2,3],
David Durden[2], Natchaya Pingintha-Durden[2], Carole Helfter[4], Eiko Nemitz[4], C. Sue B. Grimmond[5],
Janet Barlow[5], Sean Beevers[7], Gregor Stewart[7], David Dajnak[7], Ruth M. Purvis[1,6], and James D. Lee[1,6]

[1]Wolfson Atmospheric Chemistry Laboratories, Department of Chemistry, University of York, York, YO10 5DD, UK
[2]Battelle, National Ecological Observatory Network. 1685 38th Street, Boulder, CO 80301, USA
[3]Dept of Atmospheric and Oceanic Sciences, University of Wisconsin-Madison, 1225 W Dayton St, Madison, WI 53711 USA
[4]UK Centre for Ecology and Hydrology, Bush Estate, Penicuik, EH26 0QB, UK.
[5]Department of Meteorology, University of Reading, Reading, RG6 6BB, UK
[6]National Centre for Atmospheric Science, University of York, York, UK
[7]MRC Centre for Environment and Health, Imperial College London, London, UK.
[a]now at: British Antarctic Survey, Natural Environment Research Council, Cambridge, CB3 0ET, UK

**Correspondence:** Will S. Drysdale (willdrysdale@googlemail.com), James D. Lee (james.lee@york.ac.uk)

**Abstract.** During March - June 2017 emissions of nitrogen oxides were measured via eddy covariance at the British Telecom Tower in central London, UK. Through the use of a footprint model the expected emissions were simulated from the spatially resolved National Atmospheric Emissions Inventory for 2017, and compared with the measured emissions. These simulated emissions were shown to underestimate measured emissions during the day time by a factor of 1.48, but they agreed well

overnight. Furthermore, underestimations were spatially mapped and the areas around the measurement site responsible for differences in measured and simulated emissions inferred. It was observed that areas of higher traffic, such as major roads near national rail stations, showed the greatest underestimation by the simulated emissions. These discrepancies are partially attributed to a combination of the inventory not fully capturing traffic conditions in central London, and both spatial and temporal resolution of the inventory not fully describing the high heterogeneity of the urban centre. Understanding of this

underestimation may further improved with longer measurement time series ,to better understand temporal variation, and improved temporal scaling factors, to better simulate sub-annual emissions.

## 1   Introduction

Nitrogen oxides ($NO_x$), the sum of nitrogen oxide (NO) and nitrogen dioxide ($NO_2$) are air pollutants which in the urban environment are mainly emitted from anthropogenic combustion processes as NO and oxidised in the atmosphere forming $NO_2$.

$NO_2$ has been shown to exacerbate pre-existing respiratory and cardiovascular conditions (Forastiere et al., 2005). Furthermore, $NO_x$ is responsible for the formation of ground level ozone ($O_3$) in the presence of peroxy radicals (from the oxidation of volatile organic compounds) and is involved in the formation of nitrate aerosols. Tropospheric $O_3$ has been shown to cause pulmonary conditions, and has been linked to the development of asthma (McConnell et al., 2002; Saldiva et al., 2005).



London regularly faces issues with $NO_2$ concentrations, often breaching various air quality limits. $NO_x$ concentrations are
measured at a combination of sites from the Automatic Rural and Urban Network (AURN) and the London Air Quality Network
(LAQN) across the Greater London area. Average annual concentrations for the 101 sites are shown in figure 1, 57 of which
breached the European annual mean air quality limit of 40 $\mu$g m$^{-3}$ in 2017 (Council of European Union, 2008). Sites classified
as kerbside or roadside make up 51 of these, linking a lot of London's $NO_2$ issues to the transport sector.

According to the National Atmospheric Emissions Inventory (NAEI), road transport, domestic and industrial combustion are
the key sources of $NO_x$ in Greater London. Road transport is the largest single contributing sector with diesel engines receiving
much of the attention and blame for the high concentrations seen in the London. Road transport has been the target of policy
intervention in the city such as the Congestion Charging Zone (CCZ) introduced in 2003 which imposed a daily charge for
vehicles driving into the centre of London, from Monday to Friday, between 07:00 and 18:00. This policy was not intended
to improve air quality but rather, as its name suggests, congestion and also reduce $CO_2$ emissions. Very little change was
seen in $NO_x$ concentrations and at places such as Marylebone Road, a major thoroughfare which forms the northern border
of the CCZ, though increases in ambient $NO_2$ were recorded after adjusting for meteorology (Transport for London, 2016;
Grange and Carslaw, 2019). Grange and Carslaw (2019) also showed that the CCZ increased effective concentrations of $NO_2$
at Marylebone Road and they did not approach pre-CCZ levels until 2011, with the improvement of buses from Euro III to
Euro V emissions standards (5 to 2 g kWh$^{-1}$ of $NO_x$) on routes on and around Marylebone Road. Further decline was noted
with the introduction of Euro VI and hybrid buses up to 2016, where the study ended.

London's Low Emission Zone (LEZ), introduced in 2009, was aimed at improving air quality by reducing the pollution from
heavy vehicles either by reducing their number, or encouraging improved emissions control technology. This was shown to
have reduced ambient $NO_2$ levels and the number of people exposed to exceedances of the 40 $\mu$g m$^{-3}$ annual air quality limit
in several boroughs (Mudway et al., 2019).

In April 2019 London introduced the Ultra-Low Emissions Zone (ULEZ) specifically targeting vehicle emissions. The
charge applies at all times to vehicles that do not meet specific Euro classes for their vehicle type (motorbikes Euro 3, petrol
cars Euro 4, diesel cars and larger vehicles Euro 6; 0.15, 0.08 and 0.08 g km$^{-1}$ of $NO_x$ respectively) and is expected to have had
a greater impact on $NO_x$ emissions in London (Greater London Authority, 2021).

Whilst there are large amounts of ambient concentration measurements available, limited emissions measurements have been
made in London. The NAEI provides UK wide emissions estimates, and for Greater London shows that they declined from
120 to 45 ktonnes yr$^{-1}$ (62 %) between 1998 and 2017. $NO_x$ concentrations reduced between 28, 40 and 45 % at an average of
roadside, kerbside and urban background sites respectively (figure 2).

Eddy covariance (EC) measurements of NO and $NO_2$ fluxes were previously made at the British Telecom (BT) Tower
during the Clean Air for London (ClearfLo) project's intensive observation periods in 2012-13 and from an aircraft during
the Ozone Precursor Fluxes in an Urban Environment (OPFUE) campaign in 2014 (Lee et al., 2015; Vaughan et al., 2016).
During ClearfLo Lee et al. (2015) collected EC data at the BT Tower for 36 days in June - August 2012 and 28 days in March
- April 2013. These measurements suggested that the NAEI underestimated the $NO_x$ emission by a factor of 1.36 - 2.2 and was





largest for fluxes measured to the east of the tower, across all footprint distances. Diurnal profiles of $NO_x$ correlated closely with diurnal profiles of traffic flow surrounding the tower.

Airborne EC $NO_x$ fluxes were collected during 3 flights in July 2013. Vaughan et al. (2016) used these data to provide insight into the spatial change in emissions across Greater London and found the underestimation of $NO_x$ emission by the NAEI, in central London, to be similar to that found by Lee et al. (2015). The agreement between measurement and inventory improved significantly outside central London. Both of these studies also compared their results to the London Atmospheric Emissions Inventory (LAEI), an inventory which focuses on the Greater London area, and an enhancement of the LAEI using

on road emissions data, collected via remote sensing. Both of these comparisons further improved agreement and suggested that the traffic sector is responsible for much of the disagreement. The discrepancies between $NO_x$ emission measurements and inventories correlate with fleet composition in central London, where taxis and buses outnumber private vehicles (Vaughan, 2017).

   We report on EC emissions measurements of $NO_x$ from the BT Tower collected during the spring and summer of 2017. The

resulting time series is compared to the NAEI and LAEI, and supports the finding of previous studies that these inventories underestimate measured values. Additionally, these data are further developed into spatially resolved maps with the aid of footprint modelling, and we estimated the spatial distribution of these hitherto unreported $NO_x$ sources.

   In this article we will discuss the eddy covariance experimental setup, including the site, instrumentation and data processing (sections 2.1 - 2.2). We also cover in detail several sources of uncertainty in the experimental setup and provide discussion on

these with respect to the interpretation of results (section 2.2.1). In section 2.3 we cover the emissions inventories explored in this study, and the footprint modelling used to simulate an emissions time series from them. The resulting measurements are discussed in section 3.1 and their comparison with simulated emissions time series in section 3.2.

## 2   Methodology

### 2.1   Site Description and Instrumentation

Measurements of NO and $NO_2$ mixing ratios were made at the BT Tower between March - June 2017 using a closed path dual channel Air Quality Design (AQD) chemiluminescence analyser equipped with a blue-light converter for $NO_2$. The instrument is similar to those described by Lee et al. (2009) and Squires et al. (2020), and provided a sampling rate of 5 Hz. The site is a 177 m tall tower located in Central London in borough of Camden, south of Euston Road and North East of Hyde park (latitude/longitude: 51.521/-0.139 °). The surroundings are typical of the Central London area with a mixture of larger arterial

roads high traffic density and smaller side streets interconnecting them. Traffic is slow moving and *stop-start* driving conditions are common during busier periods. Surrounding buildings within a 3 km radius average ~50 m tall, with the next tallest building measuring ~130 m, placing the sampling height above the urban canopy (Environment Agency, 2015).

   A 3-D ultrasonic anemometer (Gill R3-50) was mounted on a mast atop the tower, co-located with the gas analyser sample line inlet, providing a measurement height of 190 m. The anemometer provided 3-D wind vectors and temperature derived

from the speed of sound. Air was pumped down the ~45 m sample line (PFA OD 3/8") with a target flow rate of 25 $\ell$ min⁻¹, to





the instrument which was located on the 35[th] floor. During March - June 2017 the prevailing wind direction was between west and south westerly and the median wind speed was ~6.7 m s$^{-1}$.

### 2.1.1 Instrument Calibration

NO and NO$_x$ channel sensitivities and NO$_2$ conversion efficiency were calibrated automatically every 63 hours such that data loss from calibrations was spread over the diurnal cycle. A 5.2 ppmv NO standard, traceable to the National Physical Laboratory (NPL) scale was used as a span gas and injected at 10 sccm into the sample flow. Coefficients were linearly interpolated to 1 minute resolution before being applied to the data. Both channels were zeroed for two minutes hourly using a combination of scrubbed ambient air (generated from an external Sofnofil and activated charcoal trap) and a pre-chamber zero (where O$_3$ is introduced early such that chemiluminescence occurs away from the detector). Conversion efficiency was determined via gas-phase titration of the calibration gas with O$_3$ generated from an internal mercury lamp. NO and NO$_x$ channel sensitivities were (2.4 ± 0.3) and (3.8 ± 0.6) / counts pptv$^{-1}$ respectively and conversion efficiency was (70 ± 3) % over the measurement period.

### 2.2 Eddy Covariance Calculations

Eddy covariance calculations were performed using the **eddy4R** (Metzger et al., 2017) family of R software packages, and followed the general procedure in figure A1. Calibrations were first applied as described above, and the mixing ratio and wind vector data streams joined into hourly data files. Mixing ratios were converted to mole fractions for analysis and the calculations assumed that this was dry mole fraction. In reality this was not the case due to a lack of water vapour measurement within the AQD instrument. While closed-path analysers are affected by density fluctuations from changes in temperature to a lesser extent than open-path ones, humidity can still have an effect. This effect is proportional to the concentration-flux ratio (Pattey et al., 1992) and were determined to be much less than 1 % for these measurements, in line with other NO$_x$ measurements made in a similar experimental setup (Squires et al., 2020). Fluxes were aggregated over hourly periods, and those with less than 90 % data coverage for each period were discarded. Hourly periods were used over the more traditional half hourly EC aggregation period due to the height of the measurement tower. At 190 m, lower frequency turbulence will have a greater contribution to the flux, so a longer aggregation reduces losses from these. Additionally, scaling factors for the emissions inventories were only available to hourly resolution, so there would be no analytical gain from a higher resolution time series. Spikes were removed from the data using the median filter approach described by Brock (1986) and Starkenburg et al. (2016). Subsequently, the lag between the sonic anemometer's and the AQD instrument's measurements (introduced by the spatial separation of the receptors) was corrected using high-pass filtered maximisation of the cross-correlation maximisation (Hartmann et al., 2018). Double coordinate rotation was performed to align the v wind vector with the mean flow, and reduce the average vertical wind to zero. The fluctuating components were calculated as the deviation from a linear trend calculated per hour. After the calculation of covariances, stationarity tests after Foken and Wichura (1996) were performed, and random and systematic errors calculated after Mann and Lenschow (1994). Errors presented in this manuscript are the quadratic combination of these two errors. Data was finally flagged using eddy4R's quality control scheme, which produces a quality flag based upon a combination of input





data validation, stationarity and integrated turbulence characteristics (Smith and Metzger, 2013). This resulted in 1556 hours
of high quality fluxes (66 % coverage) for the measurement period.

### 2.2.1    Additional Uncertainties

#### 2.2.1.1    Vertical Flux Divergence

EC provides measurements of local flux at the receptor. These are related, but not identical, to the surface flux. This surface
flux is what is comparable to the emissions inventories. The local flux can diverge from the surface flux due to the vertical
separation. Turbulence properties are not uniform vertically through the boundary layer; as the top of the boundary layer is
approached (the entrainment zone) vertical turbulent transport is reduced, turbulence properties are more disconnected from
the surface and the applicability of EC is diminished. This results in a vertical gradient of the turbulent flux, vertical flux
divergence. This also results in concentration enhancements below the measurement height, causing a gradient throughout the
boundary layer and is described as storage flux. The flux not registered by the receptor can be estimated from either of these
perspectives; from the rate of change in concentration with height (i.e. storage) or from proportionality with the entrainment
height (i.e. vertical flux divergence). In the case of measurements made at 190 m above the surface, the measurement height is
an appreciable proportion of the boundary layer height depending on the time of day and meteorological conditions. To account
for this we apply a correction that assumes linear divergence of the vertical flux as a function of effective measurement height
and effective entrainment height (equation 1) (Deardorff, 1974; Sorbjan, 2006; Metzger et al., 2012).

$$F' = \frac{F}{1 - \frac{z_m}{z_i}} \qquad (1)$$

Where:

- $F$ is the flux prior to correction

- $F'$ is the flux following correction

- $z_m$ is the measurement height, 190 m

- $z_i$ is the entrainment height, defined as 80 % of the boundary layer height

Modelled boundary layer height data from ERA5 were used in the determination of the correction factor (Copernicus Climate
Change Service Climate Data Store (CDS), 2017). As $z_i \rightarrow z_m$; $F' \rightarrow \infty$ so this correction was limited to boundary layer height
values greater than $2 \cdot z_m$, measurements where the boundary layer height falls below this were discarded. This divergence has
been assessed via this method due to the lack of gradient measurements available at the tower and the single point correction
as used by Squires et al. (2020) was not applied, as there was no appreciable difference between the corrected and uncorrected
fluxes. This is more likely due to attenuation of the concentration enrichments at this measurement height, rather than the lack
of stored flux.





In figure 7 the effect of the correction is shown, increasing the average diurnal profile between 23 % and 62 %. This correction has two issues which make it problematic to apply. The first is that the vertical flux divergence is strongly dependant

on boundary layer height (figure A2) which has been determined from model rather than measurements. Modelled boundary layer height was also obtained for the period 2012-01-06 to 2012-02-09 where measured boundary layer height is available for central London from the ClearfLo campaign (Bohnenstengel et al., 2015). These data had a Pearson correlation of 0.59 and an orthogonal regression of modelled vs measured gave a slope of 0.52 and an intercept of 245 m. This poor correlation and significant offset suggest a large of uncertainty in the correction factor determined from the modelled boundary layer height.

Secondly, to date no other EC measurements made at the BT Tower have applied a similar correction, which would lead to this data set being incomparable. Because of these we choose to not apply the correction beyond its presentation in figure 7. We suggest future experiments at this site consider the determination of this storage term in more detail.

### 2.2.1.2 Night-time Stationarity

When flagging data for quality control, the stationarity criterion is more readily violated when the magnitude of the calculated

flux is lower. Stationarity is considered violated if the flux calculated for a subsection of the aggregation period deviates from the flux calculated for the whole aggregation period by a predefined fraction (~30 %) (Foken and Wichura, 1996). For this reason it is more likely for the flux calculated for a subsection of an aggregation period to deviate from the whole the smaller the total flux for that period is, skewing the data set towards larger fluxes.

In figure 3-A this is shown to be the case, with the percentage of records flagged by the quality control routine rising sharply

once the magnitude of the flux fall below 10 mg m$^{-2}$ h$^{-1}$. Furthermore, as $NO_x$ emission followed a strong diurnal profile, the lower nighttime values are flagged more regularly, as seen in figure 3-B. By removing these flagged data, there is risk that the resulting values are biased high, especially at night, when stable atmospheric stratification is more likely to occur.

To quantify the effect of removing the values, the diurnal profile for $NO_x$ flux was calculated twice in figure 3-C. The black trace removes all data that has been flagged by the quality control routines and the red has only removed flagged points where

the magnitude of the flux exceeded a 5 mg m$^{-2}$ h$^{-1}$ threshold. A slight high bias was observed when the stationarity criterion was not limited by flux magnitude, and this bias was greatest at night up to ~20 %. For this analysis, all data points flagged for non-stationarity have been removed, but this bias should be considered during interpretation.

### 2.2.1.3 Sample Line Turbulence and High Frequency Corrections

Turbulent flow through the sampling line is a pre-requisite for EC measurements. Laminar flow in the sample line causes the

gas which interacts with the tubing wall to flow slower than that in the centre of the line, meaning that air parcels contain asynchronous samples, primarily causing high frequency losses (Aubinet et al., 2012; Leuning and King, 1992). Reynolds number (Re) is a quantity which is used to quantify turbulent flow of a fluid. While the transition is not well defined, Aubinet et al. (2012) suggest a Re value of < 2100 to be laminar, and > 3000 to be turbulent. More generally smaller values of Re produce laminar flow, and larger values produce turbulent flow. During the measurements at the BT Tower, flow rates in the





sample line varied between 26.7 and 2.8 $\ell$ min$^{-1}$ due to the line's particle filter becoming blocked. The filter was only irregularly replaced as access to the inlet location was limited. The Reynolds number was calculated as equation 2 and ranged between 120 and 2300. This leads to periods of time where the sample line was under a transitional or laminar regime. We observe a dependence of the measured flux on Re for values < 1500 (figure 4).

$$Re = \frac{\rho v d}{\mu} \tag{2}$$

where:

- $Re$ is the Reynolds number

- $\rho$ is the density of air, calculated at the sample line pressure and temperature, kg m$^{-3}$

- $\upsilon$ is the transit speed of the air down the sample line, m s$^{-1}$

- d is the internal diameter of the sample line, 0.00638 m

- $\mu$ is the absolute viscosity of air, calculated here as the Sutherland viscosity (Sutherland, 1893)

In figure 4 the relationship of Reynolds number with raw NO and NO$_x$ fluxes is presented. The fitting of the loess smoothed line on the binned data reveals a dependence of flux on Reynolds numbers below 1500. However, while there is this trend, there is still variability in the data during these times, so no correction has been applied, but it should be borne in mind that measured fluxes are underestimates due to this loss. The flux loss due to lack of turbulence in the sample line will primarily 195 relate to the high frequency component of the measurement, and some quantification of these is discussed below.

Due to its height the high frequency contributions to fluxes measured at the BT Tower are expected to be small, with (Helfter et al., 2016) noting that > 70 % of flux can be captured using an instrument running at 1 Hz. It was therefore expected that 5 Hz measurements would capture significantly more. Indeed, from figure 5 the 5 Hz and 20 Hz temperature co-spectra agree well, (though the 5 Hz temperature shows some signs of aliasing above 1 Hz) suggesting that the primary source of high frequency 200 loss in the measurement system is from sample line attenuation. To quantify this, high frequency correction factors for NO and NO$_2$ fluxes were calculated by comparing the normalised NO and NO$_2$ co-spectra with the normalised 5 Hz temperature co-spectra. Data for these co-spectra were selected during periods where sensible heat flux was greater than 100 W m$^{-2}$, u* greater than 0.2 m s$^{-1}$ and NO flux greater than 5 mg m$^{-2}$ s$^{-1}$ and split in to four groups by wind speed (4 - 6 and 8+ m s$^{-1}$) and median Reynolds number (< 1500 and > 2000). Data were bound into three hour chunks and low frequency damped by 205 removing 10 minute block averages to improve agreement in this region of the spectrum. Data were normalised using factors calculated between 10$^{-3}$ and 10$^{-2.5}$ Hz and then were averaged into logarithmically equally spaced frequency bins. The ratio of $Co(w'NO')$ and $Co(w'T.5Hz')$ in each bin was calculated, and the transfer functions fitted as a Lorentzian to these ratios vs frequency. The percentage difference of the area under the corrected and uncorrected co-spectra where then compared, giving correction factors of ca. 5 % across all groups. As this correction factor is small, it has not been applied to the data presented 210 here.





### 2.3 Emissions Inventories and Footprint Modelling

#### 2.3.1 National Atmospheric Emissions Inventory

The NAEI is an annual emissions estimate for a variety of species in the UK from 1970 to present. Commissioned by the Department for Environment, Food and Rural Affairs, it is currently produced by Ricardo Energy & Environment; and used
to report to European Union and United Nations greenhouse gas and air pollutant monitoring programmes (Defra and BEIS, licenced under the Open Government Licence (OGL), Crown Copyright 2020, 2017; Council of European Union, 2016). Primarily the inventory provides total emissions estimates, required by these monitoring programmes. Calculations assimilate activity data and emissions factors from a wide range of sources and combines them to form an emission. Emissions are categorised into the 11 source sectors defined by the Selected Nomenclature for sources of Air Pollutants (SNAP) along with
point sources (table A1) (European Environment Agency, 2016).

Once emissions estimates as a whole are compiled, the emissions are gridded using spatial information relevant to the SNAP sector. For example road transport uses road network location, local fleet composition from automatic licence plate recognition statistics and the annual average daily flow of traffic (Tsagatakis et al., 2018). Combined with emissions factor and activity data this provides a 1 km$^2$ resolution map of emission in the UK. The 2017 version of the inventory is used in this work.

#### 2.3.2 London Atmospheric Emissions Inventory

The LAEI is an annual emissions that has been produced periodically since 2006 covering; spanning Greater London at a 1 km$^2$ resolution and covering also covers a wide range of air pollutants. It is commissioned and published by Transport for London and the Greater London Authority, with the most recent version built for 2016 (the version used in this work).

Four source sectors are included in the LAEI - Transport, Industrial and Commercial, Domestic, and miscellaneous. A
notable difference here is the grouping of commercial sources with industrial, where in the NAEI they are grouped with domestic sources. The inventory used in this work was provided with hour of day scaling for the transport sector, but otherwise has been treated the same as the NAEI.

#### 2.3.3 Footprint Modelling and Simulated Emissions Estimates from Inventories

To link the measured fluxes to the surface, we used the 2-D footprint model by Kljun et al. (2004) with an additional cross
wind component by Metzger et al. (2012). This produced a footprint at 100 m × 100 m resolution per hour of flux data, using meteorology statistics from the eddy covariance calculations, supplemented with modeled boundary layer height data from ERA5 (Copernicus Climate Change Service Climate Data Store (CDS), 2017) (boundary layer height is not a strong predictor in the footprint model, so the issues highlighted in section 2.2.1.3 are not of concern here) and a surface roughness length of 1.1 m (the average within 5 km of the BT Tower, Drew et al. (2013)). The footprint consists of a grid of these 100 x 100 m
cells, each with an associated weighting of that area's contribution to the measured flux, where the sum of the weights equals 1. The footprints were trimmed to 90 % of the total footprint weights i.e cells containing weights in the 10[th] percentile and



below are removed, as above this threshold the footprint area grows rapidly, and the individual contribution from each grid cell is diminished. The average footprint for the measurement period can be found in figure 6, overlaid on a map of the 4 main sectors which contribute to the NAEI within the footprint area.

These hourly footprints were used to simulate an emissions time series from the spatially resolved NAEI for 2017 and LAEI for 2016. This was achieved by first extracting, on a by sector basis, the inventory's grid cell (1 $km^2$) values at the centre of each hourly footprint's (0.1 $km^2$) grid cells. Each of these extracted values is weighted by that cells contribution to the total hourly footprint and finally summing over all grid cells within the footprint. Each sector is then scaled to the month of year, day of week and hour of day through the use of a selection of anthropogenic emissions profiles (figures A3 and A4) (Coleman et al.,

2001; van der Gon et al., 2011; Brookes et al., 2013). These scaled sectors can then be summed to produce a total simulated emission as would be observed at the BT Tower. The same method was applied to the LAEI, except in-built hourly scaling for the road transport layer was used instead of the scaling factor. Day of week and month of year factors were still applied.

    The footprints were also used to map the measured and expected emissions spatially. This was achieved using the *polarPlot()* function from the **openair** R package (Carslaw and Ropkins, 2012). This function traditionally bins a scalar (often pollutant

concentration) by wind speed and direction, and produces an interpolated surface via gam smoothing (Wood, 2017). Along-wind distance to the footprint maxima was provided to the function in the place of wind speed, resulting the output's radial axis having the units of meters. This could then be overlaid on a map. The along wind distance to the footprint maxima is a simple method to produce these surface maps and neglects some of the information gained by the use of a 2-D footprint model. More sophisticated methods of producing footprint topographies (Mauder et al., 2008; Kohnert et al., 2017) may allow for a more

direct comparison with the inventories, but are out of scope of this study.

## 3   Results and Discussion

### 3.1   Measurements

In figure 7, the time series of $NO_x$ concentration and flux are shown, along with average traffic volume from a selection of automatic traffic counters within the footprint of the tower (Transport for London, 2018), and modelled boundary layer height

from ERA5; 0.25 x 0.25 ° global meteorology product; ECMWF ReAnalysis 5, Copernicus Climate Change Service Climate Data Store (CDS) (2017). Median $NO_x$ concentrations showed two peaks at 08:00 h (24.81 ppbv) and 21:00 h (19.5 ppbv) (all times presented herein are Coordinated Universal Time, UTC). There was a local minimum between the peaks of 13.27 ppbv at 18:00 and the lowest median concentration was overnight at 03:00 h (8.74 ppbv). The decrease during the day is primarily due to dilution by a growing boundary layer; as the boundary layer grows the volume into which the pollutant is emitted increases.

This results in the same emission being unable to sustain as high a concentration. This is important to note as these morning and evening peaks in concentration can easily but erroneously be ascribed to rush hour activity. Indeed, from the measurements of $NO_x$ flux, it can be observed that emissions remain reasonably constant during the day. The median ($\pm$ total error) diurnal profile of $NO_x$ flux showed a steep rise in emission from ($4.71 \pm 1.14$) to ($18.67 \pm 4.96$) mg $m^{-2}$ $h^{-1}$ between 04:00 h and 08:00 h, and remained between 17.88 and 20.91 mg $m^{-2}$ $h^{-1}$ until 18:00 h at which point it gently declined to between 3.66 and





5.53 mg m$^{-2}$ h$^{-1}$ overnight by 23:00 h. The day time average (between 08:00 and 19:59) was (18.19 $\pm$ 4.86) mg m$^{-2}$ h$^{-1}$, (19.78 $\pm$ 5.33) mg m$^{-2}$ h$^{-1}$ on weekdays and (16.01 $\pm$ 3.97) mg m$^{-2}$ h$^{-1}$ on weekends.

### 3.2   Comparison with Inventories

Comparison of these measurements with emissions inventory was performed by generating a simulated emission time series via the method described in section 2.3.3. This method transforms the annual values in each emissions inventory into an
hourly time series. It should be acknowledged that much of this temporal upscaling was achieved using general anthropogenic emission scaling factors not associated with the inventories directly. Here we discuss both the temporal and spatial performance of these simulated emissions time series against measurement. As the magnitudes of both the NAEI and LAEI emissions are similar, these results are discussed in terms of the NAEI until they are broken down by emissions sector, in which case both inventories are presented.

Figure 8 compares the diurnal profiles of the measured and simulated NAEI emissions. Across all wind sectors the measured emissions are higher during the day (08:00 - 18:00 h) by a factor of 1.48. Overnight (23:00 - 04:00 h) the measured and simulated emissions agree well, with a ratio of 1.02 .Removing the diurnal profile (as this is imposed in the inventory by the scaling factors), the daily median measured value was 1.29 $\times$ the simulated (13.65 vs 10.56 mg m$^{-2}$ h$^{-1}$).

By wind sector the story is more varied. The north and east show the measurements spiking significantly above the simulated
emissions during the morning, but then showing good agreement throughout the rest of the day, again with the simulated emissions being higher at night. This is reflected in the daily medians of both of these sectors being much closer to unity (ratios of 1.13 and 0.99 respectively). In the south and west the day time underestimation by the inventory can be observed (ratios of 1.54 and 1.53 respectively), whereas overnight the agreement is better than the overall average at 1.03 and 1.00 respectively. Table 1 presents the sector breakdown from the simulated emissions time series to explore whether any particular sector may
be responsible for the missing emissions. When just considering the south, the large contribution from the traffic sector might suggest that there is a systemic underestimation here. However, the traffic sector contribution to the west is similar to that from the north and east, where the agreement with inventory is much better.

In figure 9 the diurnal profiles have been separated by day of week. Here the simulated emission has been presented by hourly bars, separated by source sector, and has additionally been presented alongside the average traffic volume measured at
24 automatic traffic counting sites, selected from those that occupied grid cells making up the first 80 % of the contribution to the flux footprint climatology. The primary diurnal variation in the simulated emissions comes from the road transport and combustion sectors, and here can been see to be driving the double peak during the day. Diurnally, the road transport sector follows the measured traffic volumes, as this is driven by diurnal scaling factor and for the weekday/weekend, the difference is again driven by the day of week scaling factor. The effect of the latter can be seen more clearly in figure 10 where day of
week averages of both measured and simulated emissions are presented. Here, decreased agreement between the simulated and measured emissions is seen on Saturday, primarily being driven by the scaling factors decreasing at the weekend (figures A3 and A4). From the measurements, Saturday's emissions are much more comparable to the weekdays than Sunday. This may be a property more unique of London. Much of the combustion within the footprint climatology is from *commercial* combustion





**Table 1.** Inventory sector contribution to simulated emissions, by wind sector.

| Inventory | Sector | % Contribution to Simulated Emission | | | | |
| | | North | East | South | West | Total |
| --- | --- | --- | --- | --- | --- | --- |
| NAEI | Road Transport | 29.91 | 32.43 | 41.52 | 29.42 | 35.30 |
| | Domestic Combustion | 20.50 | 45.40 | 21.74 | 15.97 | 20.16 |
| | Industrial Combustion | 8.55 | 16.23 | 18.90 | 5.01 | 12.73 |
| | Other Transport | 3.82 | 5.97 | 4.92 | 3.66 | 4.61 |
| | Other | 8.54 | 6.77 | 3.14 | 3.37 | 3.73 |
| LAEI | Road Transport | 39.68 | 48.46 | 52.96 | 42.53 | 47.86 |
| | Domestic Combustion | 2.49 | 1.78 | 2.11 | 2.67 | 2.09 |
| | Industrial Combustion | 24.50 | 48.91 | 30.63 | 21.99 | 26.38 |
| | Other | 0.00 | 0.00 | 0.00 | 0.00 | 0.00 |

- as demonstrated by the domestic combustion sector being significantly diminished in the LAEI which groups commercial
combustion into the industrial combustion sector. Commercial sources would be expected to continue activity on a Saturday,
which coupled with limited reductions in average traffic flow, could explain why this behaviour may be more unique to central
London and therefore is less well captured by general scaling factors.

A *budget closure* style exercise was also conducted to probe the effect of different degrees of scaling have on the measure-
ment to inventory ratio. This was done by constructing the simulated inventory time series as previously described along with
a second series where only the monthly scaling factors were applied. Both of these, along with the measurement time series,
were averaged by month. The assumption being that the hourly or weekly scaling factors should be mostly averaged out over
the course of a month, as they sum to unity over 24 hours or 7 days respectively. This assumption is perturbed by the uneven
sampling across the diurnal profile, with there being ~3 times more missing values overnight, driven by processes such as
stationarity violations (section 2.2.1.2). This is reflected in the comparison in table 2 where only using monthly factors leads to
underestimates that are generally larger than when all scaling factors are used, driven by the day time scaling factors increasing
the values, without their nighttime counterparts. These underestimates (1.37 - 1.84 for the NAEI and 1.35 - 1.73 for the LAEI)
are similar to the day time underestimates presented earlier, reinforcing the idea that there are missing sources of $NO_x$ within
the flux footprint climatology surrounding the BT Tower and not simply an artefact from the emissions simulation method.

To further explore the differences wind sector dependence of the emissions and begin to identify potential missing sources,
figure 11 shows daytime and nighttime average emissions by 22.5 ° wind direction bins. The daytime underestimation is
sustained through the west and south direction, whereas the underestimation that manifested as a morning peak in the north
and east diurnals is much narrower. It can also be seen that to the north west the simulated emissions agree or even slightly
over estimate versus the measurements. Overnight the agreement improves in all directions.





**Table 2.** Ratio of Measured to simulated $NO_x$ emission from both the NAEI and the LAEI. For each inventory the both cases of all scaling factors and only monthly scaling factors are shown

|  | LAEI | | NAEI | |
| --- | --- | --- | --- | --- |
| Month | Month Only | All Scale | Month Only | All Scale |
| 3 | 1.68 | 1.35 | 1.52 | 1.37 |
| 4 | 1.49 | 1.33 | 1.53 | 1.37 |
| 5 | 1.73 | 1.60 | 1.83 | 1.52 |
| 6 | 1.58 | 1.70 | 1.84 | 1.60 |

In figure 12, the surface mapping approach as described in section 2.3.3, has been applied to the measured and simulated (NAEI) emissions in an attempt to further elucidate the spatial discrepancies observed here. The measured flux is mapped in panel A and the simulated inventory time series in panel B, then difference between these is shown in panel C. Areas to the north-east, south and west that have been highlighted by the measurements as significant sources, are not captured by the inventory in this treatment, revealing sources that are not fully resolved by the inventory. This method of mapping these emissions cannot be easily validated, and by using just along wind distance to footprint maxima, much of the spatial information is collapsed - however, the same caveats apply to both the measured and simulated maps and comparison of them may provide some insight into their respective differences.

The measured emissions (figure 12-A) place the emission enhancements to the south-west over the areas of Oxford Street and Regents Street and the enhancement in the north-east is over Euston Station and Marylebone Road. Both of these areas are busy with road transport, and Euston Station also has a bus depot which filters into the already congested Marylebone road. It is possible that these enhancements are not well captured by the NAEI as they are localised at features much finer than the inventories spatial resolution (shown by the lack of similar structure in figure 12-B), and they could also be due to specific driving conditions found on these roads, which will not be well described by a bottom up approach of inventory construction. The decreased emissions over The Regent's Park to the north-west are again likely due to the resolution of the inventory - though the map being sensitive to this large green space within the footprint provides some qualitative validation of the method.

## 4 Conclusions

During March - June 2017 $NO_x$ flux was measured at the BT Tower in central London via eddy covariance. A footprint model was used to simulate an emissions time series from the spatially resolved NAEI and LAEI. We show that while the BT Tower site continues to be able to produce high quality fluxes, there are uncertainties still to be addressed, namely that of vertical flux divergence/storage of emissions below the receptor location. These uncertainties primarily affect the upper bound of the emissions measurement.





The inventories underestimated (1.48 ×) the measured $NO_x$ emissions during the day, but showed improved agreement overnight. This underestimation was present in monthly averaged comparisons also (1.35 - 1.84 ×). Using the footprint model again, spatial differences in the measured and simulated emissions were explored and using this method it appeared particularly
congested regions around the tower were not well represented by the inventory.

It is clear from these measurements that there are contributions to the $NO_x$ emission in central London not captured by the inventory, however, they do not allow us to untangle their sources explicitly. While this is currently the longest time series of measured $NO_x$ emission in the city, 3 - 4 months of data necessitates the use of scaling factors to make comparisons with the inventories. The monthly averaged comparison shows that the underestimation is not a factor of this process alone but the
day of week comparison shows flaws in the use of these factors that are generalised across anthropogenic activity - that the activity of central London may not be reflected precisely by them. Collection of a time series spanning greater than 12 months would allow for annual budgets to be compared (and ongoing data collection will provide this in future work). Although, a difference between the measured and inventory emissions is likely to persist, a single annual data point will do little to untangle from where this discrepancy arises. So in many cases, the ability to produce high temporal resolution emissions estimates will
still be necessary, and provision of this information from inventory constructors would improve the comparisons that can be made. Indeed the LAEI used here provided hourly scaling for the road transport sector - but this does not yet provide sufficient information when it is mixed with the other required factors.

Resolving the measurements spatially does provide hints as to where the discrepancies may be found, here we showed that the highest emissions around the tower were close to locations that experience high congestion. The change in traffic
emissions due to congestion is not something that is directly parameterised in the bottom up inventories used here, but further investigation may be able too close the gap between them and measurement.

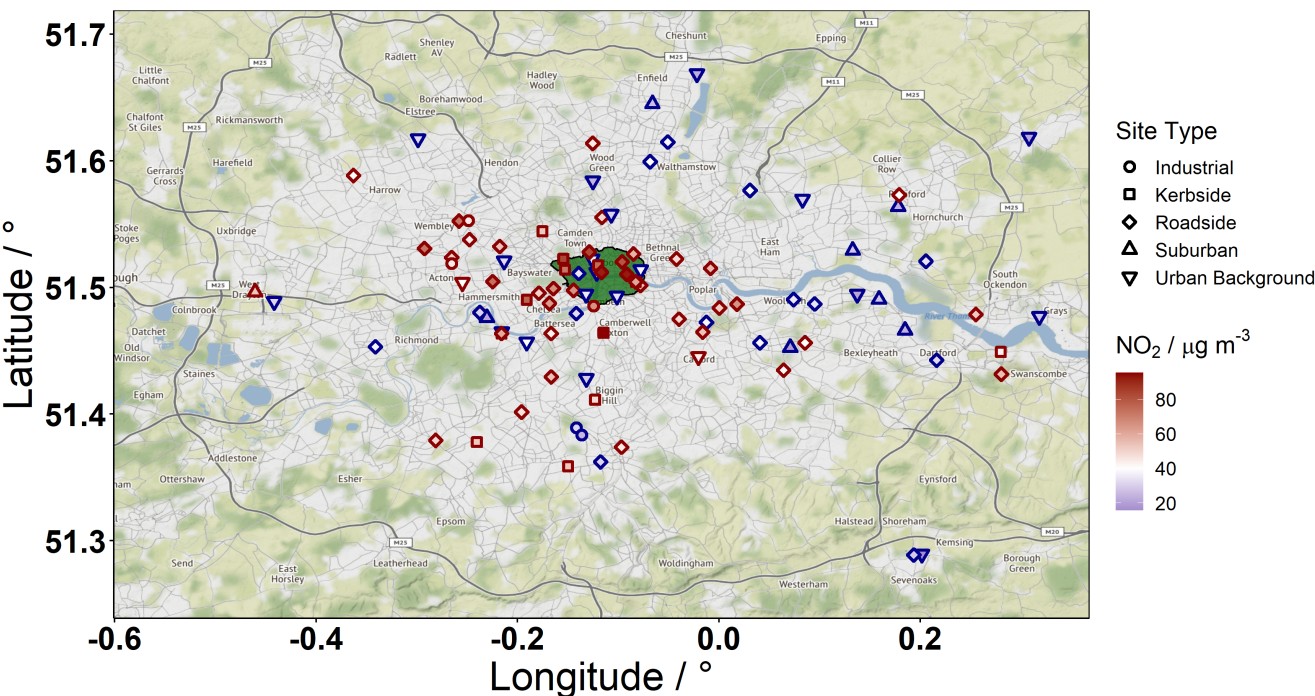

**Figure 1.** 101 air quality monitoring sites located in and around Greater London. Sites are coloured by their annual mean $NO_2$ concentration for 2017 ($\mu$g m$^{-3}$). Point shape denotes the type of measurement site. Point borders change from blue to red above the 40 $\mu$g m$^{-3}$ air quality limit. 57 sites had annual mean concentrations above this limit in 2017. The area which encompasses the congestion charging zone and ultra low emissions zone is shown in green. Map tiles by Stamen Design, under CC BY 3.0. Data ©OpenStreetMap contributors 2021. Distributed under the Open Data Commons Open Database License (ODbL) v1.0. Tiles accessed via the **ggmap** R package (Kahle and Wickham, 2013).



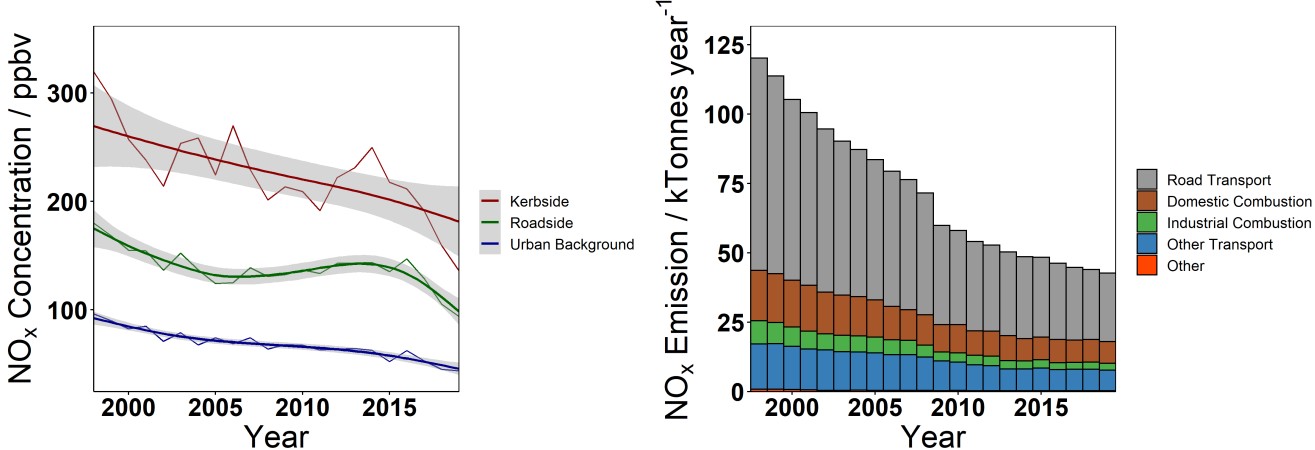

**Figure 2.** Left - Change in average concentration at roadside, kerbside and background sites in Greater London between 1998 and 2019. All sites with available data were first annually averaged, followed by the grand mean of all sites of a given type per year. A gam model was fit to the data to produce the smooth line, shading shows the standard error in this fit. Right - NAEI emissions for Greater London. As historical spatially resolved versions of the NAEI are not available, these data were generated by scaling each sector of the spatially resolved inventory for 2019 within the Greater London area by their relative value in the historical UK total emissions data. This assumes that London has generally followed the UK trend in $NO_x$ emissions.

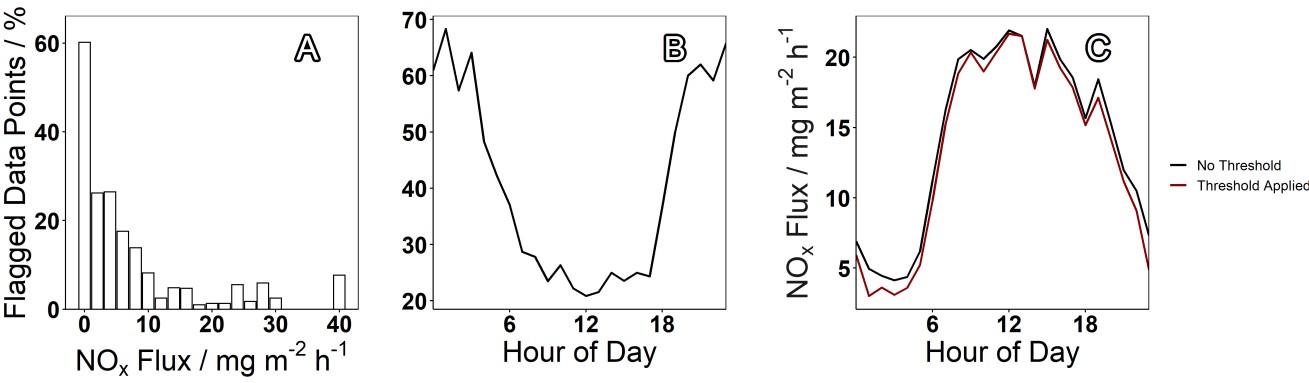

**Figure 3.** A - Percentage of flux records flagged by quality control routines in 2 mg m$^{-2}$ h$^{-1}$ bins. B - Percentage of flux records flagged by quality control routines by hour of day. C - NO$_x$ flux diurnal cycle where black has had all records flagged by quality control routines removed and red has only had them removed if the flux magnitude was also greater than 5 mg m$^{-2}$ h$^{-1}$



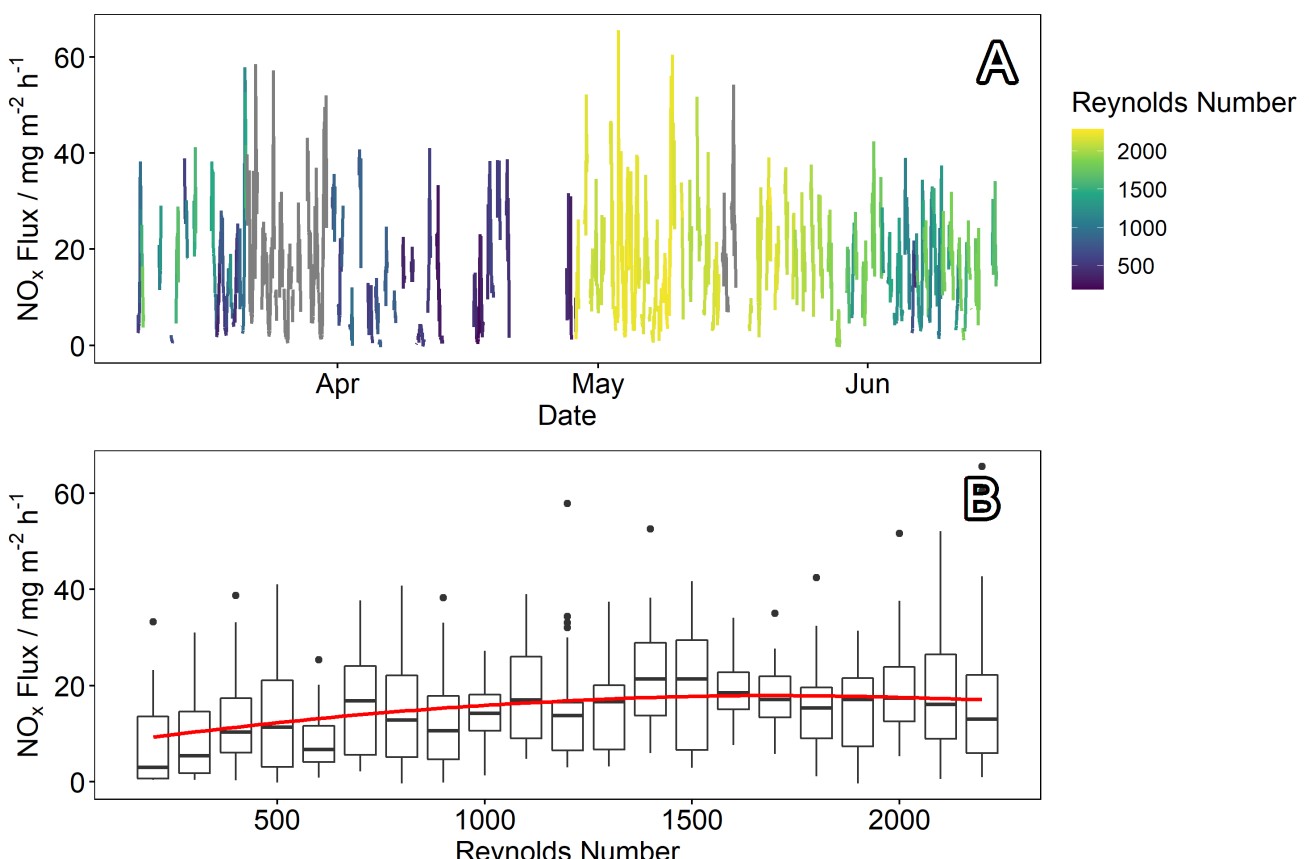

**Figure 4.** A - Unfiltered $NO_x$ flux coloured by Reynolds number. Grey periods are where sample flow data is unavailable. B - $NO_x$ flux against binned Reynolds number (bin width 100). Boxes show median value as the horizontal bar and $25^{th}$ and $75^{th}$ percentile at the limits of the box. Whiskers extend to 1.5 × the inter quartile range, data that fall outside of this range are plotted as points. A loess smoothed fit shows increasing dependency of $NO_x$ flux on Reynolds numbers below 1500



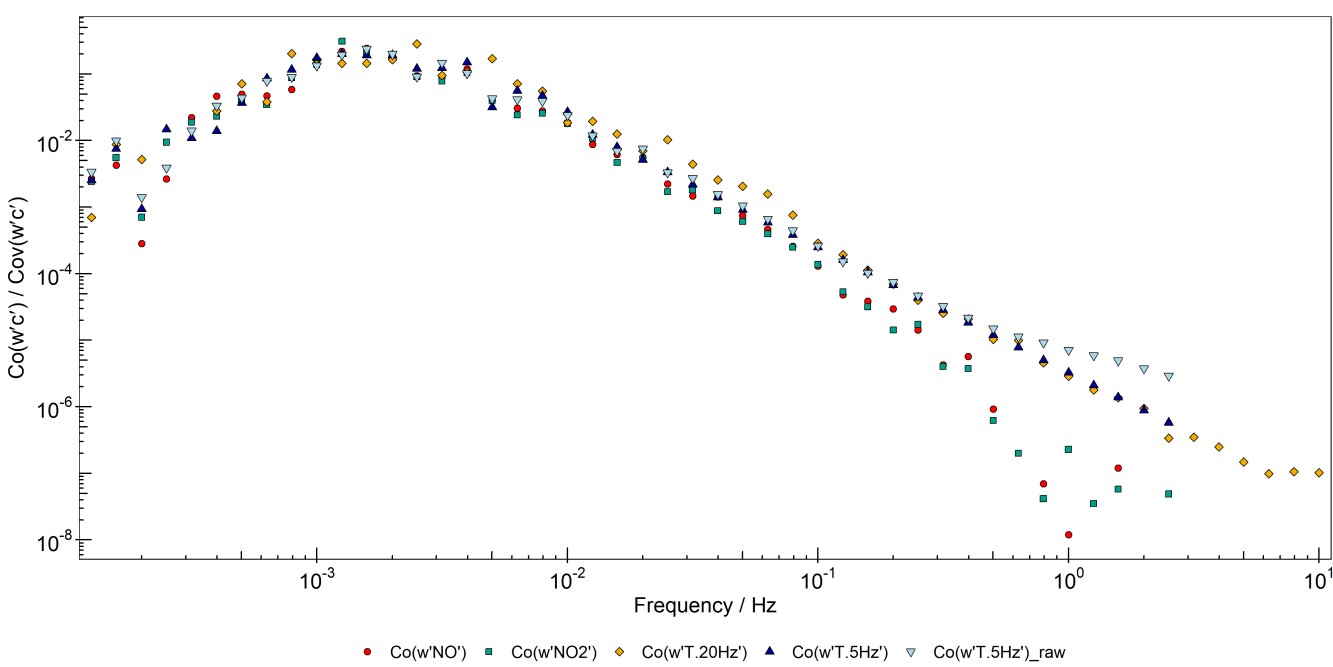

**Figure 5.** Co-spectra of vertical wind with NO (red, circle), NO$_2$ (green, square) and three versions of temperature. Temperature was sampled from the same anemometer, but data was recorded at 20 Hz (yellow, diamond and 5 Hz (light blue, down-triangle, "raw"). The raw 5 Hz temperature exhibits some aliasing at the higher frequencies, which was corrected for (dark blue, up-triangle) by fitting a linear model over the $10^{-2}$ to $10^{-1}$ frequency range and replacing values a frequencies > $10^{-2}$ with this model.

**Figure 6.** The sum of the NAEI layers corresponding to SNAP sectors 07, 02, 03, and 08 (see table A1) to show the spatial distribution of the majority of NO$_x$ emission in central London. The 30, 60 and 90 % contributions to the flux footprint climatology for EC measurements made between March - July 2017 are shown in white. The red point shows the location of the BT Tower. Map tiles by Stamen Design, under CC BY 3.0. Data ©OpenStreetMap contributors 2021. Distributed under the Open Data Commons Open Database License (ODbL) v1.0. Tiles accessed via the **ggmap** R package (Kahle and Wickham, 2013).



**Figure 7.** Time series (left) and median diurnal profiles (right) of (top to bottom) measured of $NO_x$ flux (green solid is pre-vertical flux divergence correction, blue is including this correction), simulated emissions from the NAEI, $NO_x$ concentration, traffic volume and modelled boundary layer height. Shaded periods on time series highlight weekends. Shaded regions on diurnal profiles refer to median absolute deviation in diurnal averaging, except in the case of $NO_x$ flux, where this region represents the average total error in flux measurement.

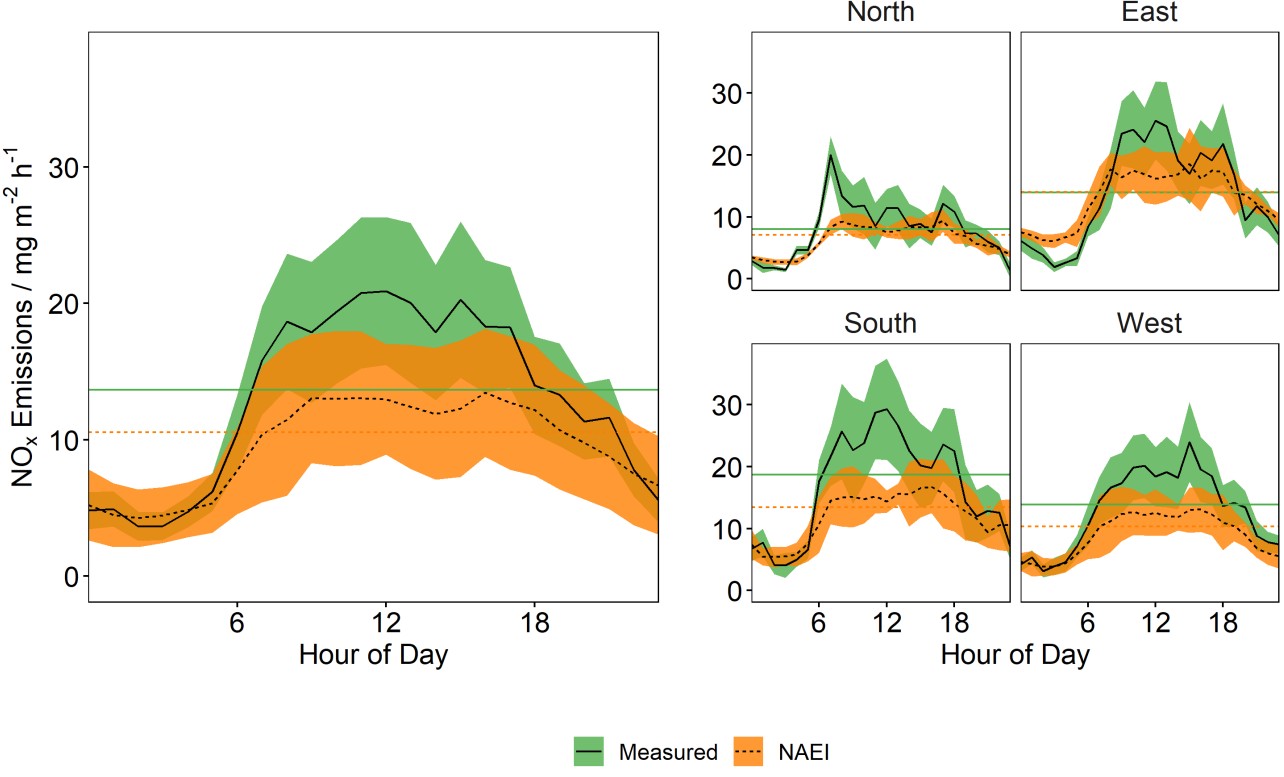

**Figure 8.** Median diurnal profiles of $NO_x$ emissions measured (green, solid) at the BT Tower March - July 2017 and simulated emissions from the NAEI (orange, dashed). Shaded region shows total (random + systematic) error in flux measurement for the measured emission and median absolute deviation in diurnal averaging for simulated emissions. Horizontal lines show the daily median values. Left hand side shows the average diurnal profiles for the total measurement period, right hand side shows this separated by wind direction.

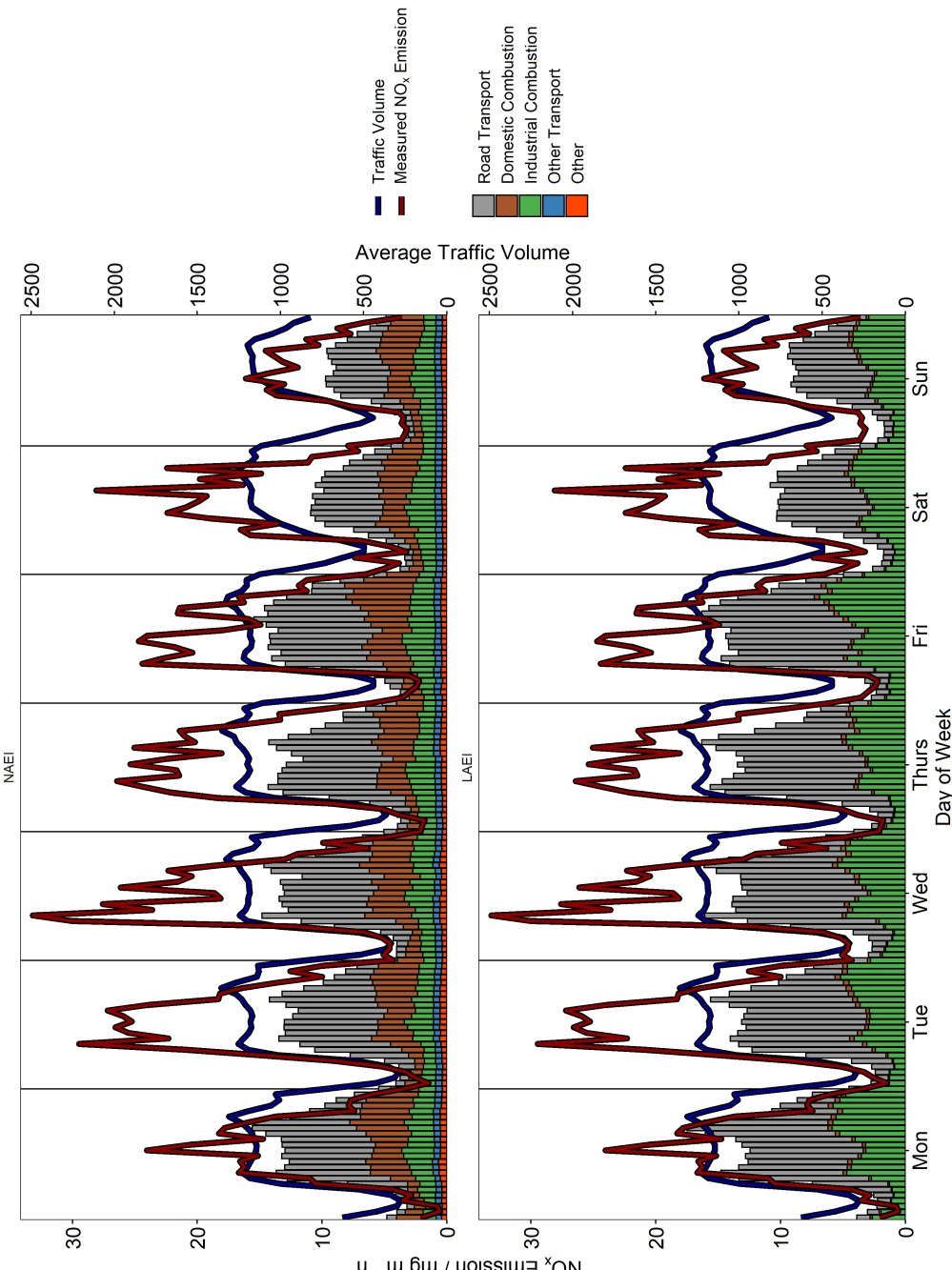

**Figure 9.** Average diurnal profiles of $NO_x$ emissions measured (red) at the BT Tower March - July 2017 and the NAEI's estimated emission (bars) from within the flux footprint, separated by day of week. NAEI emissions are coloured by source sector contribution. Median traffic volume from 24 automatic traffic counters surrounding the site are shown in blue.



**Figure 10.** Daily averaged measured NO$_x$ emissions by day of week shown as red points and stacked bars of simulated emissions coloured by source sector for the NAEI (solid colour) and LAEI (hatched).

**Figure 11.** NO$_x$ emissions (red) measured at the BT Tower March - July 2017 and the NAEI's estimated emission (bars) from within the flux footprint, averaged by 22.5 ° wind sector bins. NAEI emissions are coloured by source sector contribution. The left hand panel shows all data between 0800 - 1959 and the right hand panel shows all data between 2000 - 0759.

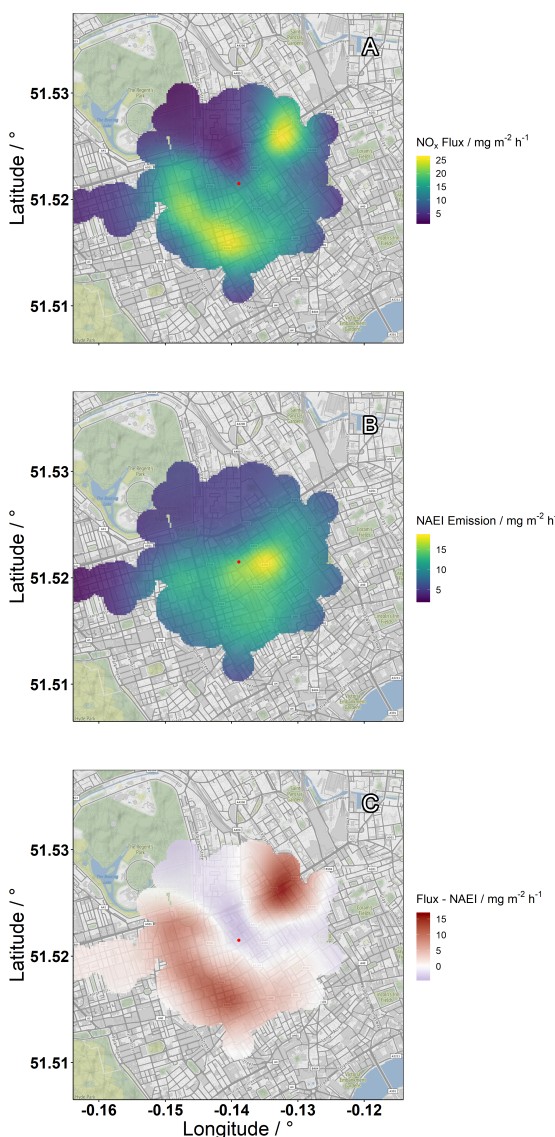

**Figure 12.** A - Measured $NO_x$ flux as a function of along-wind distance to the maximum flux contribution on the radius, separated by wind direction. B - NAEI $NO_x$ emissions estimate as a function of along-wind distance to the maximum flux contribution on the radius, separated by wind direction. C - B subtracted from A, red shows measurement greater than inventory, blue shows inventory greater than measurement. Map tiles by Stamen Design, under CC BY 3.0. Data by OpenStreetMap, under ODbL. Tiles accessed via the **ggmap** R package (Kahle and Wickham, 2013).



## 5 Code availability

The eddy4R v.0.2.0 software framework used to generate eddy-covariance flux estimates can be freely accessed at https://github.com/NEONScience/eddy4R. The eddy4R turbulence v0.0.16 and Environmental Response Functions v0.0.5 soft-

ware modules for advanced airborne data processing were accessed under Terms of Use for this study (https://www.eol.ucar.edu/content/cheesehead-code-policy-appendix) and are available upon request.

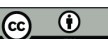



A

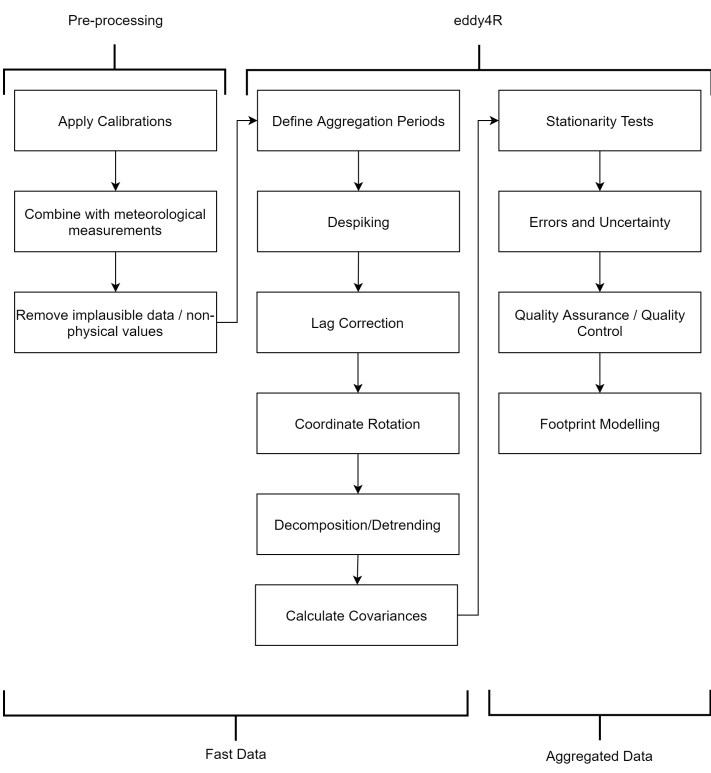

**Figure A1.** Schematic eddy covariance calculation workflow.





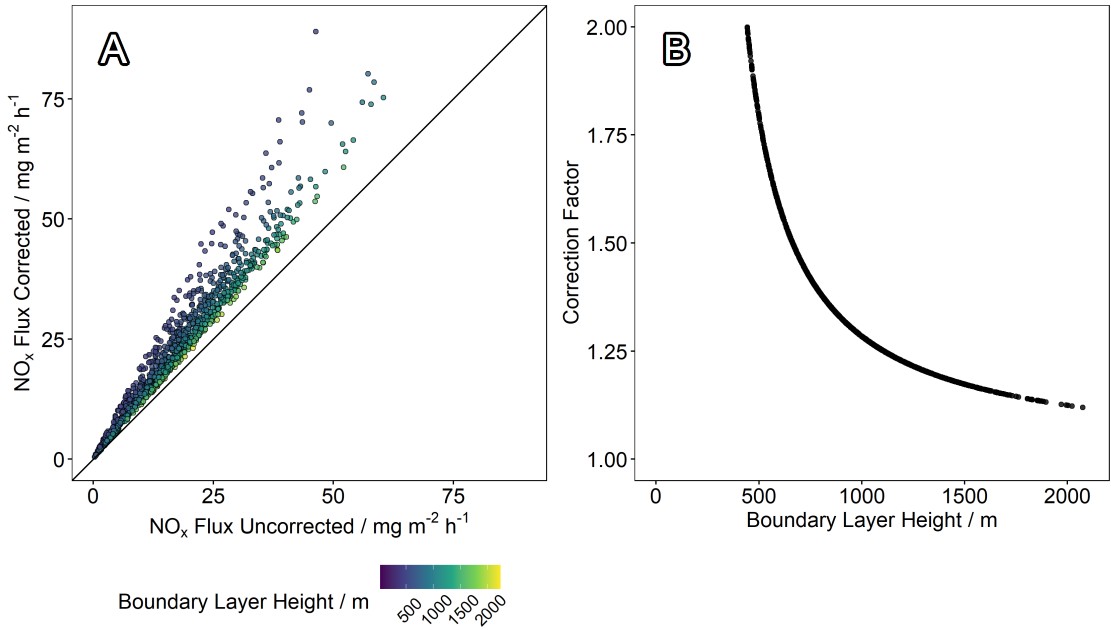

**Figure A2.** Visualisation of the vertical flux divergence corrections with respect to boundary layer height. A - Corrected $NO_x$ flux against uncorrected $NO_x$ flux, coloured by boundary layer height. B - Correction factor against boundary layer height.





**Figure A3.** Hour of day scaling factors for the four SNAP sectors (07, 02, 03, and 08, see table A1) contributing to the majority of NO$_x$ emission around the BT Tower, coloured by day of week. Overlapping points for identical profiles have been offset in the x direction to improve readability.





**Figure A4.** Month of year scaling factors for the four SNAP sectors (07, 02, 03, and 08, see table A1) contributing to the majority of NO$_x$ emission around the BT Tower, coloured by day of week. Overlapping points for identical profiles have been offset in the x direction to improve readability.





**Table A1.** Selected Nomenclature for sources of Air Pollutants sector definitions as used in the NAEI Defra and BEIS, licenced under the Open Government Licence (OGL), Crown Copyright 2020 (2017). The four sectors with the largest contribution to NO$_x$ emission within the footprint of the BT Tower are highlighted in bold

| SNAP Sector | NAEI Label | Definition |
|---|---|---|
| 01 | energyprod | Combustion in Energy and Transformation |
| 02 | **domcom** | Combustion in Commercial, Institutional, Residential and Agriculture |
| 03 | **indcom** | Combustion in Industry |
| 04 | indproc | Production Processes |
| 05 | offshore | Extraction and Distribution of Fossil Fuels |
| 06 | solvents | Solvent Use |
| 07 | **roadtrans** | Road Transport |
| 08 | **othertrans** | Other Transport and Mobile Machinery |
| 09 | waste | Waste |
| 10 | agric | Agriculture, Forestry and Landuse Change |
| 11 | nature | Nature |





*Author contributions.* WSD made $NO_x$ measurements, calculated the fluxes, performed footprint modelling and analysed the data. WSD wrote the manuscript and produced the figures, with input from co-authors. ARV, FAS, SJC, SM, DD and NPD provided support with the flux calculations and interpretation of the data. SM, DD and NPD also provided training on the eddy4R software. CH and EN provided supporting measurements from the site and aided in interpretation of the data. CSBG and JB provided meteorological data for the site. SB, GS and DD gave information on the LAEI and provided the version of the inventory used in this study. RMP and JL reviewed the manuscript and provided input on the interpretation of the data

*Competing interests.* The authors declare that they have no competing interests.

*Acknowledgements.* This work was supported by UKRI grant NE/T001917/1 for the 'Integrated Research Observation System for Clean Air project'. WSD acknowledges PhD studentship funding from the National Centre for Atmospheric Sciece, award ref: ncasstu002. The National Ecological Observatory Network is a program sponsored by the National Science Foundation and operated under cooperative agreement by Battelle. This material is based in part upon work supported by the National Science Foundation through the NEON Program. Additional thanks to Neil Mullinger for supporting the measurement infrastructure at the BT Tower. The authors would like to thank operational staff at the BT Tower for supporting this research.





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
