# Peer review of "Eddy Covariance Measurements Highlight Sources of Nitrogen Oxide Emissions Missing from Inventories for Central London"

_Atmospheric Chemistry and Physics, 2021_

## Author Response (AR1)

**Response to Reviewers Comments for: Eddy Covariance Measurements Highlight Sources of Nitrogen Oxide Emissions Missing from Inventories for Central London**

We are grateful to the two reviewers who have provided a thorough critique of this manuscript. Their comments are fair and have helped to improve the quality of the study. Both reviewers focused on two main areas:

- The spectral treatment of the data to improve uncertainty estimates at the high and low frequency ends of the measured data. We have reworked and expanded our analysis in this area and now believe it addresses the concerns raised.
- The methods used to estimate flux loss due to vertical divergence at the measurement height. Here we have added greater discussion to our methods which aims to alleviate the issues with the original draft

We provide details on the adjustments made to the manuscript for these points, and other raised, in-line below. Figure numbers refer to those in the revised manuscript.

**Anonymous Referee #1**

Referee comment on "Eddy Covariance Measurements Highlight Sources of Nitrogen Oxide Emissions Missing from Inventories for Central London" by Will S. Drysdale et al., Atmos.

The study by Drysdale et al. investigates NOx eddy covariance flux measurements in context of emission inventories in London. It presents a follow up study on previous campaigns at the same site. A main finding seems to be that NOx emissions are still underestimated by local emission inventories in London. The paper is well written and meets ACP quality criteria for research papers. I suggest publication after the comments below have been addressed adequately.

Line 85: Air was pumped through a 45 m sampling line. Obviously flowrates varied between 2.8 and 26 l/min due to clogged filters. What was the variation of delay time calculated by EddyRe through this line? The dependence of NOx flux vs Reynolds Number suggests a bias up to 50% for the first half of the measurement period. I understand that this would make the discrepancy between measured fluxes and inventory even larger (thus not change the major conclusion of the paper), but I wonder whether this large bias justifies the inclusion of the early campaign data without correction.

The sample flow rate changed the calculated time delay by around 2 s, but the lag times were allowed to drift within. We have reworked high-frequency correction analysis to further quantify the losses due to reduced sample line flow. We calculated the cospectra under 4 Reynolds number regimes and the losses derived from this were in the range of 5 - 10 % and seemingly uncorrelated with Re. Some of this lack of correlation is likely due to the limited number of spectra that fall into this category ( < 5 %). This reduces the expected change from that suggested by Figure 5 quite considerably.  We additionally calculated the average diurnal profile of NOx flux with and without data where Re < 500 (about 15 % of the full time series) and the change observed for any point was between -0.5 and 1.5 mg m$^{-2}$ h$^{-1}$. With all of these factors considered and the more detailed high

frequency analysis now presented in figure 6, we have not opted to remove the data here. We also do not apply the derived correction as it remains small. This has the added bonus of streamlining the comparison with Lee et al. 2014 which has included after other comments.

Line 100: The EC analysis is done with eddy4R aggregating the flux calculation into 60 min intervals. This is a non-standard flux averaging interval because stationarity can become an issue for longer time averaging intervals. It would be good to present a statistical measure that justifies a 60 min flux calculation interval. Why not simply apply spectral corrections as described by Massman et al. 2002 (doi: 10.1016/S0168-1923(02)00105-3) The cospectral analysis could give a good metric on flux averaging intervals - e.g. how long is long enough?

The aggregation period was selected as 60 min to match with the temporal scaling factors available for the inventory. To check that the 60 min aggregation periods were appropriate, fluxes were also calculated using a 30 min aggregation period and averaged to 60 mins, and no significant difference was observed. We have included text referring to this in lieu of conducting extra spectra analysis. Concerns regarding instationaries over the longer aggregation period should also be somewhat diminished by the use of linear detrending – already selected due to NO and NO2 rapid changes over the diurnal cycle.

Line 149: Figure 7 is mentioned before Figure 3,4,5, and 6. It also appears that Figure 5 appears before figure 4 - Figure numbering should be consistent – i.e. in sequential order.

Figure 7 is now figure 3 and all others have been incremented accordingly.

Line 135: This correction formula merits more discussion. For example what was the assumption for the concentration jump of NOx across the PBL?

Section 2.2.1.1 has been updated and expanded in relation to comments from referee 2. These improvements should address the point raised here.

Line 250: The temporal disaggregation methodology of the yearly emission inventories is not clear. Why would you still scale the LAEI hourly emission data by week and month?

This difference in scaling was specific to the transport sector, which was provided with hour of day scaling built in. We have updated these sentences to make this difference more clear.

Line 253: It is not clear what the open R package is really used for. The 2D flux footprint should already give you an appropriate weighing function that can be applied to bottom up fluxes. Using the along wind distance to the footprint maximum in conjunction with a polarPlot function seems an unnecessary (and rather semiquantitative) step here.

Initially the 2D footprint weighting matrices were used to produce these maps – they are more complicated owing to the spatial and temporal heterogeneity of the emissions surrounding the tower – so more work is required before they are ready to present. The method applied here condenses the spatial information from the 2D footprint and draws attention to broad features of the dataset and allows for a simple mapping approach. We agree that this method is far from quantitative and have adjusted the language describing the method to make this clear.

Section 3.1: NOx fluxes are reported in mg/m2/h – in my opinion molar units would be much more appropriate, since NOx is the sum of two species with different molecular weight. Reporting fluxes in mg/m2/h leads to an important loss of information. This is particularly relevant since most of NOx is emitted as NO from combustion processes. I therefore highly recommend to change from mass to molar units as is done for mixing ratios, which are all reported as ppbv and not ug/m3.

The NO and NO2 fluxes were calculated separately in molar units before conversion into mass units and combination into NOx flux. This should preserve the information whilst avoiding the conversion of the inventories into molar units, where speciation is unknown. We have added brief detail about this to section 2.2.

Line 273: It would be informative if the authors expanded their discussion here, comparing their results to NOx flux measurements elsewhere and previous studies at the location. Are NOx fluxes in London quite a bit higher or lower than in other urban areas? e.g. Marr et al., 2013: doi: 10.1021/es303150y; Karl et al., 2017: doi:

10.1038/s41598-017-02699-9; Guidolotti et al., 2016: doi:

10.1016/j.agrformet.2016.11.004; Squires et al., 2020: doi: 10.5194/acp-20-8737-2020; How similar or different are the results to Lee et al.,2015 (doi: 10.1021/es5049072), who published NOx fluxes at the same location? Have fluxes changed since then or rather stayed constant?

Section 3.3 and figure 13 have been added summarising this literature and directly comparing this study with Lee et al. 2015.

Line 280: what is meant be temporal upscaling?

Temporal upscaling refers to the process being used to increase the temporal resolution of the annual inventory to hourly values via the scaling factors.
* * *
**Anonymous Referee #2**

Referee comment on "Eddy Covariance Measurements Highlight Sources of Nitrogen Oxide Emissions Missing from Inventories for Central London" by Will S. Drysdale et al., Atmos.

Chem. Phys. Discuss., https://doi.org/10.5194/acp-2021-982-RC2, 2022

The authors present eddy covariance measurements of NOx fluxes over London and interpret the results in terms of predominant sources and bottom-up predictions. The topic is of relevance to ACP and in general the quality of analysis is appropriate. I list some comments below that in my view should be addressed prior to acceptance.

===========================

Main comments

===========================

Section 2.2. Here the authors derive a 23-62% correction due to vertical flux divergence but then don't apply it as too uncertain due to issues with the model mixing height. What helps put the study on more solid footing is that this correction would only increase the measured fluxes, and the authors are already inferring an inventory underestimate. However a couple aspects of this are somewhat surprising, at least to me, and merit more discussion.

- The first is the size of the effect, and here it would help to give some information on the modeled boundary layer height values. Based on Eq 1, 23-62% corrections require heights of only be 500-1000m. If I'm reading it correctly, the observational comparison later in this section indicates that the modeled boundary layer heights are biased substantially low, so that the flux divergence influence as estimated is too big.

- I believe application of Eq 1 implies that the measurement height is always out of the constant flux layer, and further that there isn't actually a constant flux layer at all ... i.e. the correction begins at the surface rather than at the top of the surface layer. Some more physical justification is needed of its applicability.

These are both excellent points regarding our application of a vertical flux divergence correction. We have taken these on board and reworked our correction based on the two points: a correction of the observed model offset in 2012 should be considered for the 2017 data, and the correction should only begin applying to zm above the constant flux layer. This latter point also has the advantage that when the tower is in the constant flux layer, no correction need be applied. The tower is much more frequently in this constant flux layer when the BLH is corrected. Section 2.2.1.1 now includes this in its discussion. Figure A2 has been changed to visualise the how these additional terms change the correction. (they both make it considerably smaller). We still opt not to apply the correction as the boundary layer height uncertainty still demonstrates strong influence over the result (for example can we confidently apply a correction factor from 2012 on this 2017 data). The upside is that if we take accept the method used here for vertical flux divergence is improved, the correction factors we are not applying are much smaller, reducing overall uncertainty in the measured emissions.

- Another surprising aspect is the inconsistency between the entrainment-based correction approach (23-62%) and the single-point correction approach (0%). The authors attribute the latter to "attenuation of the concentration enrichments at this measurement height, rather than the lack of stored flux" but it is not clear to me how physically plausible this actually is. Some more information / discussion is needed.

With the reduction in magnitude of the correction, the disagreement between the single point approach and this divergence approach is less stark. Nevertheless, we have expanded the discussion on why this is, and primary argue that the taller the tower a single point at the top of the air column is less representative of that below it, and as such will be less able to provide a correction and as such is "attenuated". The top-down approach we apply here is more well constrained by the use of boundary layer height.

2.2.1.3 The authors derive a 5% spectral correction due to high-frequency sampling losses. However there was large variability in sampling flow (3-30 L/min) and turbulent characteristics (Re 120-2300). Does the degree of high-frequency loss vary significantly across these conditions? Is it really only 5% for the low-flow / laminar periods?

We have expanded the high frequency analysis in response to this and comments from referee 1. See the above discussion which we also believe addresses this comment.

Sampling took place on a 13m mast atop a 177m building. Surrounding buildings are all quite a lot shorter but to what degree might the BT Tower itself disrupt the sampled flow field in a way that would bias the fluxes?

Wind tunnel studies have shown a deflection on the order of ~3 degrees into the updrafts measured at the tower (10.1016/j.jweia.2011.05.001). However, we are not able to comment on how these

directly impact the EC measurements made. It could be expected that some of this deflection would be compensated for by the double coordinate rotation performed on the wind vectors as a part of the eddy covariance processing. Some tests were conducted using wind sector based planar fit but do significant differences were observed vs double coordinate rotation. This is something we will consider exploring using longer time series currently being collected, but is out of the scope of this study.

2.2 I appreciate the discussion of eddy covariance QA/QC. It would also be helpful to show ogives and to give information about the range of sampling lag times.

Some info on lag times has been added. A description of the tests done on 60 min and 30 min aggregation periods has also been added – but as these showed no significant distance ogives were not calculated.

2.3.1    please state the temporal resolution of NAEI in this section. Is it annual?
2.3.2    what is the spatial resolution of LAEI? Also 1km2?

Both of these are stated within their respective sections

L252, "Day of week and month of year factors were still applied". Unless I missed it, this is the first mention of temporal variability in the inventories. Such factors need to be more carefully described in the corresponding methods sections.

This sentence has been revised for clarity

Conclusions. It seems that pinning down the uncertainty due to entrainment / storage should be an important priority for future work, as this term is similar in magnitude to the inventory bias inferred. Are there plans along these lines that can be mentioned in the conclusions?

Conclusions. The reader is left a bit unsatisfied by the lack of take home messages. The authors might consider discussing some implications of their findings; e.g. what do the derived emission errors mean for AQ predictions?

Conclusions have been edited and expanded upon following the response to both referees, highlighting the areas for future work with additional data gathered at the BT Tower, and also some colour for putting the results of this work into context with policy and activity changes we are seeing currently.

===========================

Technical/editorial comments

===========================

some typos and grammatical issues throughout; please correct.

Lines 25-35, the narrative here as written is confusing and hard to follow.

Some changes made to readability, and some additional sentences added for context.

67, "hitherto unreported sources" implies missing sources when it seems the problem is mainly underestimation of known sources. Perhaps "underreported"

Noted and amended

Figures are not referred to in order

Figure 7 is now figure 3 and all others have been incremented accordingly.

Table 1, I'm confused here because some directions the sectors add up to <<100% and others add to >100%. Please clarify what is going on.

There was a bug in the table generation, this has now been fixed, along with the text discussing it.

---

## Author Response (AR2)

We are grateful for this final review – here we have fixed the incorrect reference noted by the reviewer.

Additionally, we noticed a type on line 214. Previously this line read "…sensible heat flux was > 5 W m$^{-2}$…", this has been corrected to "…sensible heat flux was > 50 W m$^{-2}$…"